# The impact of Astragaloside IV on the inflammatory response and gut microbiota in cases of acute lung injury is examined through the utilization of the PI3K/AKT/mTOR pathway

**Cheng Luo[1], Yuanhang Ye[1], Anqi Lv[1], Wanzhao Zuo[1], Yi Yang[2], Cheng Jiang[2], Jia Ke[2]\***

**1** Clinical College of traditional Chinese Medicine, Hubei University of Chinese Medicine, Wuhan, China, **2** Hubei Provincial Hospital of Traditional Chinese Medicine, Hubei Academy of Traditional Chinese Medicine, Affiliated Hospital of Hubei University of Traditional Chinese Medicine, Wuhan, China

\* kejia@hbhtcm.com

## Abstract

### Objectives

Astragaloside IV (AS-IV) is a natural triterpenoid saponin compound with a variety of pharmacological effects, and several studies have clarified its anti-inflammatory effects, which may make it an effective alternative treatment against inflammation. In the study, we aimed to investigate whether AS-IV could attenuate the inflammatory response to acute lung injury and its mechanisms.

### Methods

Different doses of AS-IV (20mg·kg$^{-1}$, 40mg·kg$^{-1}$, and 80mg·kg$^{-1}$) were administered to the ALI rat model, followed by collection of serum and broncho alveolar lavage fluid (BALF) for examination of the inflammatory response, and HE staining of the lung and colon tissues, and interpretation of the potential molecular mechanisms by quantitative real-time PCR (qRT-PCR), Western blotting (WB). In addition, fecal samples from ALI rats were collected and analyzed by 16S rRNA sequencing.

### Results

AS-IV decreased the levels of TNF-α, IL-6, and IL-1β in serum and BALF of mice with Acute lung injury (ALI). Lung and colon histopathology confirmed that AS-IV alleviated inflammatory infiltration, tissue edema, and structural changes. qRT-PCR and WB showed that AS-IV mainly improved inflammation by inhibiting the expression of PI3K, AKT and mTOR mRNA, and improved the disorder of intestinal microflora by increasing the number of beneficial bacteria and reducing the number of harmful bacteria.

**Data Availability Statement:** All relevant data are within the manuscript and its Supporting information files.

**Funding:** National Administration of Traditional Chinese Medicine Inheritance Studio Construction Project, Education Letter of the State Office of Traditional Chinese Medicine (2022) No. 245; Natural Science Foundation of Hubei Province, 2023AFD173.

**Competing interests:** The authors declare that they have no known competing financial interests or personal relationships that could have appeared to influence the work reported in this paper.

## Conclusion

AS-IV reduces the expression of inflammatory factors by inhibiting the PI3K/AKT/mTOR pathway and optimizes the composition of the gut microflora in AIL rats.

## Introduction

ALI, a frequently occurring respiratory ailment, possesses a substantial morbidity and mortality rate [1]. The cause may be direct damage to the lungs, such as viral or bacterial infections, inhalation or inhalation of harmful substances. In addition, indirect causes such as sepsis, severe trauma, and reperfusion injury may also lead to injury [2]. Initially, ALI is distinguished by inflammation in the lungs and disorders in the permeability of microvessels, ultimately leading to the development of acute respiratory distress syndrome, with a clinical mortality rate ranging from 35–55% [3, 4]. Despite significant advancements in the treatment of ALI through mechanical ventilation and oral corticosteroids, the mortality rate remains noteworthy [5]. When pathogenic bacteria invade the lungs, bacterial fragments and products stimulate the body to release inflammatory mediators that bind to pulmonary epithelial related receptors and promote epithelial cell apoptosis, causing damage to the alveolar and respiratory mucosa [6]. It is therefore necessary to further explore effective alternative therapies to combat inflammation.

The human gastrointestinal tract is inhabited by microbiota, which is commonly referred to as gut microbiota [7]. In recent years, through extensive research on intestinal microbiota, scientists have discovered a reciprocal regulatory function between the lung and intestinal microbiota, known as the gut-lung axis [8]. It has emerged as a popular subject among researchers. Recent studies [9, 10] have linked the microbiome of the digestive system to various respiratory conditions, such as ALI. Disruption of gut microbiota can cause intestinal mucosal damage and inflammatory reactions, and the intestine is connected to the lungs through pathways such as the gut-lung axis. Pathogenic bacteria can migrate to the lungs through this way, causing lung injury. Intestinal and pulmonary infections caused by imbalance of gut microbiota lead to mucosal tissue damage, inducing inflammation and immune response, which is the key to pathogenesis. The gut-lung axis plays an important role in ALI caused by gut microbiota [11]. It has been hypothesized that the microbiota in the gut plays a role in controlling both immune and immunological responses. In recent studies, enteric bacteria *Enterobacteriaceae* were found in the lungs of ARDS patients without systemic infection, indicating that gut microbiota can migrate through the gut-lung axis and alter the lung microbiota, affecting lung health [12, 13]. In the ALI model, alterations in the composition and variety of the intestinal microbiota contribute to the restoration of ALI and the decrease in inflammation [11]. During lung injury, the gut microbiota will accumulate in the lungs, leading to an increase in levels of inflammatory factors such as IL-6 and inducing pulmonary inflammation, thereby affecting the progression of ALI/ARDS and closely related to the prognosis of the disease [14]. Hence, the regulation of gut microbiota could potentially serve as a viable treatment choice for ALI.

An extract known as AS-IV was derived from Astragalus mongholicus Bunge, a naturally occurring substance. This information was published online at http://www.worldfloraonline. org/taxon/wfo-0000185128. Accessed on: 24 Aug 2023 [15]. This compound is a triterpenoid saponin with properties that regulate the immune system, reduce inflammation, and act as an antioxidant [16], commonly employing for the treatment of cancer, inflammation, as well as

bacterial and viral infections [17, 18]. Toxicological studies have found that AS-IV is relatively safe and does not show significant side effects [19]. Previous studies have shown that AS-IV exerts its protective effect on the lungs by improving lung ventilation function, reducing alveolar epithelial and capillary permeability, and inhibiting pulmonary inflammation in sepsis induced ALI rats [20]. Recent research suggests that AS-IV affects the gut microbiota composition and the production of butyrate [21, 22]. Additionally, AV-IV has demonstrated its ability to safeguard against lung injury caused by PM2.5 by activating the PI3K/AKT/mTOR signaling pathway [23]. Nevertheless, the exact mechanism of action not be elucidated. Based on the preceding discourse and previously conducted studies, a distinct correlation exists between ALI, gut microbiome, and AS-IV. This research created a rat model of ALI and used 16S rRNA gene sequencing to examine the exact mechanism through which AS-IV treats ALI by means of the gut microbiota, demonstrating a clear correlation between ALI, gut microbiome, and AS-IV. To offer a basis for AS-IV treatment of ALI based on science.

## Materials and methods

### Animals

Male Sprague-Dawley rats weighing 180-200g were bought from the animal research facility at Three Gorges University in Hubei, China. The rats were obtained under the certificate number SCXK (E) 2022–0012. The experimental animal center of Hubei University of traditional Chinese medicine provided housing for all animals. The animals were kept at a temperature of 22°C±2°C, a relative humidity of 60%±10%, and followed a 12-hour light/12-hour dark cycle. They were also given unrestricted access to standard food and water. To minimize the effects of excessive stress, experimental modeling began following a period of adaptive feeding lasting 7 days. This study was approved by the Experimental Animal Ethics Association of Hubei University of Traditional Chinese Medicine (HUCMS00302536).

### Plant material

The purchase of AS-IV was made from Beijing Solaibao Technology Co., LTD located in Beijing, China. The batch number is 84687-43-4 and its molecular formula is $C_{41}H_{68}O_{14}$ with a molecular weight of 784.98. The HPLC percentage is equal to or greater than 98% (Fig 1). Before the experiment, a suitable quantity of AS-IV powder was dissolved in a solution of 0.5% sodium carboxymethyl cellulose (CMC) to prepare the AS-IV low-dose group at a dosage of $20mg·kg^{-1}$, the AS-IV medium-dose group at a dosage of $40mg·kg^{-1}$, and the AS-IV high-dose group at a dosage of $80mg·kg^{-1}$. Dexamethason (Solarbio,50-02-2); LPS (Solarbio,1031S032).

### ALI model and treatment

In the ALI experiment, rats were first anesthetized with isoflurane gas to relieve pain or discomfort, and then a certain amount of LPS dissolved in normal saline ($5mg·kg^{-1}$,100 μL/ rat) was injected into the trachea. Create the model after success, thirty rats were randomly assigned to five groups, and six normal feeding rats served as the Control group. Total 6 groups with six rats in each group. The groups were named as follows: Control (blank group), ALI (model group), DEX (dexamethasone group; $1.0mg·kg^{-1}$), L-AS-IV (AS-IV low-dose group; $20mg·kg^{-1}$), M-AS-IV (AS-IV medium-dose group; $40 mg·kg^{-1}$), and H-AS-IV (AS-IV high-dose group; $80mg·kg^{-1}$) [24, 25]. 12 hours after modeling, the same volume of ordinary saline was orally administered to the rats in the Control group and the ALI group every day. The DEX group received an intraperitoneal injection of $1.0mg·kg^{-1}$ dexamethasone. Groups of AS-IV was given different doses of AS-IV by gavage. All groups were treated for 7 days, twice/

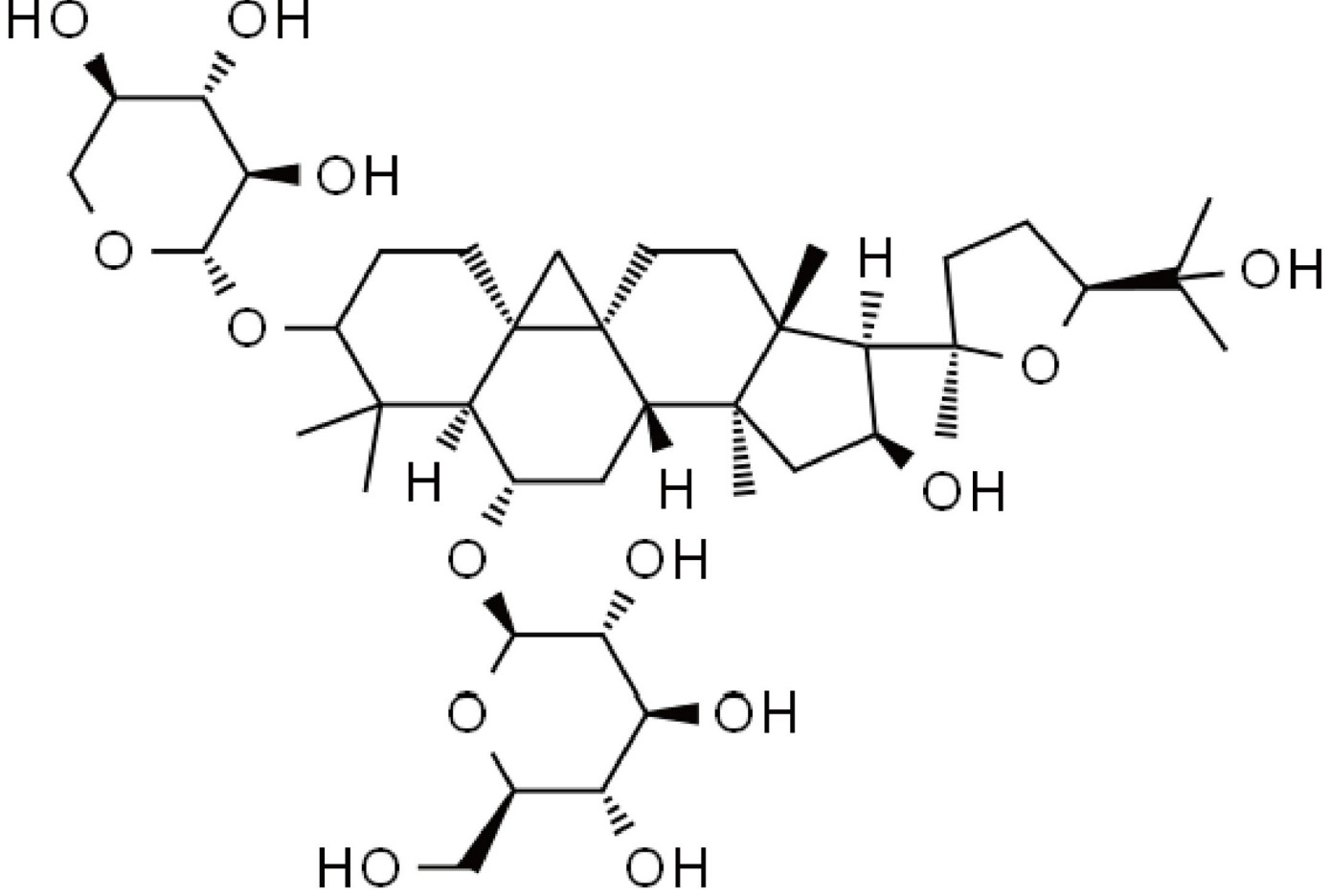

**Fig 1. AS-IV molecular formula.**

d, once in the morning and once in the evening. All rats were euthanized and place the rat separately in a small room and gradually add a lethal dose of $CO_2$ concentration (e.g., 33% administered after 1 minute) to make them fall asleep. Then, let them die by cervical dislocation. Serum, BALF, lung tissue, colon tissue, and stool samples were collected.

## Histological analysis

The colon and lung tissues were immediately preserved in 4% formalin at room temperature (RT) for one night, then embedded in paraffin wax and stained with H&E.

## Measurement of inflammatory cytokines

The inflammatory factors were assessed by measuring the serum and BALF levels of IL-1β (CSB-E08056r), IL-6 (CSR-E04642r), and TNF-α(CSB-E11986r) in rats from each group using the ESLEA kit (CUSABIO, Wuhan, China) as per the provided instructions.

## Real-time quantitative PCR

Total RNA was immediately extracted using Trizol reagent (Aidlab, 252250AX) from rats lung tissue. First-strand cDNA synthesis was carried out using the PrimeScriptTM RT-PCR Kit

**Table 1. Primer sequences.**

| Name | Primer | Sequence | Size |
|------|--------|----------|------|
| Rat GAPDH | Forward | 5'– CCTCGTCCCGTAGACAAAATG –3' | 133bp |
|  | Reverse | 5'– TGAGGTCAATGAAGGGGTCGT –3' |  |
| Rat PI3K | Forward | 5'– CCTTGTCGTCTCACCTTC –3' | 158bp |
|  | Reverse | 5'– CCATCCTCAGTTCACCATT –3' |  |
| Rat mTOR | Forward | 5'– GCCTCTGCTCTCCTTAATG–3' | 153bp |
|  | Reverse | 5'– CGTCAACTCCAACTTCTAC –3' |  |
| Rat AKT | Forward | 5'– CTGACTTCCTCTTCCTGTAG –3' | 198bp |
|  | Reverse | 5'– ACCTCTTGGCATACTTGAC –3' |  |

(Takara,RR014).The SYBR Green PCR Master Mix (VAZYME, Q111-02) was used to assess the mRNA of PI3K, AKT, and mTOR on an iCyler iQ Real-Time PCR System (Bio-Rad Laboratories Inc., USA)according to the manufacturer's guidelines. The reaction conditions included an initial step at 94˚C for 2 minutes, followed by 40 cycles at 94˚C for 10 seconds, 60˚C for 1 minute, and 72˚C for 30 seconds. Real-time PCR primers were acquired from Servicebio Biotechnology Co., Ltd., located in China. The quantification results were compared to GAPDH as a reference mRNA by utilizing the $2^{-\Delta\Delta Ct}$method. The primers for the target gene can be found in Table 1.

## Western blot analysis

30 mg lung tissues were homogenized in the RIPA lysis buffer (Solarbio, China). Then, the tissue or cell proteins were obtained via centrifuging at 12,000 rpm at 4˚C for 25 min. After centrifugation, the supernatant was collected, and a BCA protein assay kit was used to measure the protein content. Take 20μL of the proteins were separated on SDS-PAGE gel and transferred onto a polyvinylidene fluoride membrane. The membranes were blocked with 5% nonfat milk at RT for 1 hour and were cut horizontally to incubate the corresponding antibodies of different target proteins. Then, the membranes were incubated at 4˚C overnight with the following antibodies: Rat anti-PI3K (1:1000, Rat#abs119725, Absin), Rat anti-p-PI3K (1:1000, Rat#AF3241, Affinity), Rat anti-AKT (1:1000, Rat#4961, CST), Rat anti-p-AKT (1:1000, Rat#4961, CST), Rat anti-mTOR (1:1000, Rat#ab32028, abcam), Rat anti-p-mTOR (1:1000, Rat#ab32028, abcam), and Rat anti-β-ACTIN (1:1000, Rat#ab8245, abcam). Following three rounds of TBST washing, the secondary antibodies conjugated with HRP-Goat anti Rat (1:3000, Rat#E-AB-1003, Elabscience) were incubated at room temperature for a duration of 2 hours. After TBST cleaning again for 3 times, the protein bands were detected using ECL hypersensitive luminescent agent (Solarbio, China), and ImageJ 7.0 software (NIH, America) was used to analyze the gray level of each strip, and the relative expression levels of each strip gray level/β- ACTIN gray level were identified as PI3K, p-PI3K, AKT, p-AKT, mTOR, and p-mTOR proteins.

## 16S rRNA for gut microbiota

Fecal samples were quickly collected from rat cecum and frozen with liquid nitrogen and stored at -80˚C. DNA extraction and 16S rRNA gene sequencing of fecal samples were conducted with the assistance of Wuhan Billion Huineng Biotechnology Co., Ltd. (China). Total genomic DNA samples were extracted using the OMEGA Soil DNA Kit (M5635-02) (Omega Bio-Tek, Norcross, GA, USA), following the manufacturer's instructions, and stored at -20˚C prior to further analysis. The quantity and quality of extracted DNAs were measured using a

NanoDrop NC2000 spectrophotometer (Thermo Fisher Scientific, Waltham, MA, USA) and agarose gel electrophoresis, respectively. Specific amplification of the 16S rRNA gene was performed. PCR amplification of the bacterial 16S rRNA genes V3–V4 region was performed using the forward primer 338F (5'-ACTCCTACGGGAGGCAGCA-3') and the reverse primer 806R (5'-GGACTACHVGGGTWTCTAAT-3'). PCR amplicons were purified with Vazyme VAHTSTM DNA Clean Beads (Vazyme, Nanjing, China) and quantified using the Quant-iT PicoGreen dsDNA Assay Kit (Invitrogen, arlsbad, CA, USA). After the individual quantification step, amplicons were pooled in equal amounts, and pair-end 2 250 bp sequencing was performed using the Illlumina MiSeq platform with MiSeq Reagent Kit v3 at Bioyi Biotechnology Co.,Ltd.(Wuhan, China).

## Statistical analysis

The SPSS 21.0 software was utilized for conducting statistical analysis. In cases where the data followed a normal distribution and the variances were homogeneous, a one-way analysis of variance (ANOVA) was applied to compare multiple groups. To compare between groups, the minimum significant difference (LSD) test was applied. The results were expressed as the mean ± standard deviation ($\bar{x}$±s). The rank sum test was utilized in cases where the data did not adhere to normality or homogeneity of variance, and the outcome was represented by the median (25%,75%) [M(p25,p75)]. A p-value<0.05 suggests that the observed difference is statistically significant.

## Results

### AS-IV alleviates LPS-induced lung and colon injury

In order to clarify the pathological alterations of ALI rats in every group, H&E staining was employed to evaluate the abnormalities in lung and colon histopathology. The Control group showed no alteration in lung histopathology, with a distinct and intact alveolar structure, normal alveolar spacing, and absence of inflammatory cell infiltration, bleeding, and edema in the alveolar and pulmonary interstitium. The ALI group exhibited disruption of the alveolar architecture, widening of the alveolar space, and edema, bleeding, as well as significant infiltration of inflammatory cells in the alveolar and pulmonary interstitium. The changes were significantly reduced by AS-IV and DEX treatments, with H-AS-IV treatment showing the most notable impact (Fig 2A). The Control group exhibited no alterations in the colonic histopathology, displaying a clear and condensed structure of all layers, intact mucosal epithelium, plentiful and arranged intestinal glands. Conversely, the ALI group presented colonic tissue characterized by mucosal congestion, structural impairment, inflammation resulting in injury, edema in the mucosa and submucosa, significant inflammation due to cell infiltration, and glandular necrosis. AS-IV and DEX treatment significantly attenuated these changes, with H-AS-IV having the most significant effect (Fig 2B).

### AS-IV suppresses the expression of inflammatory factors in LPS-induced ALI rats

To explore the mechanism by which AS-IV alleviates the inflammatory infiltration in lung and colon tissues of LPS-induced ALI rats, we detected the expression of IL-6, IL-1β and TNF-α in serum and BALF, as shown in Fig 3A–3F. The results showed that compared with the Control group, the levels of inflammatory factors TNF-α, IL-6 and IL-1β in serum and BALF of rats in ALI group were significantly increased, with statistical significance (*P < 0.01*). Compared with ALI group, the levels of IL-6, IL-1β and TNF-α in serum and BALF of rats in H-AS-IV and

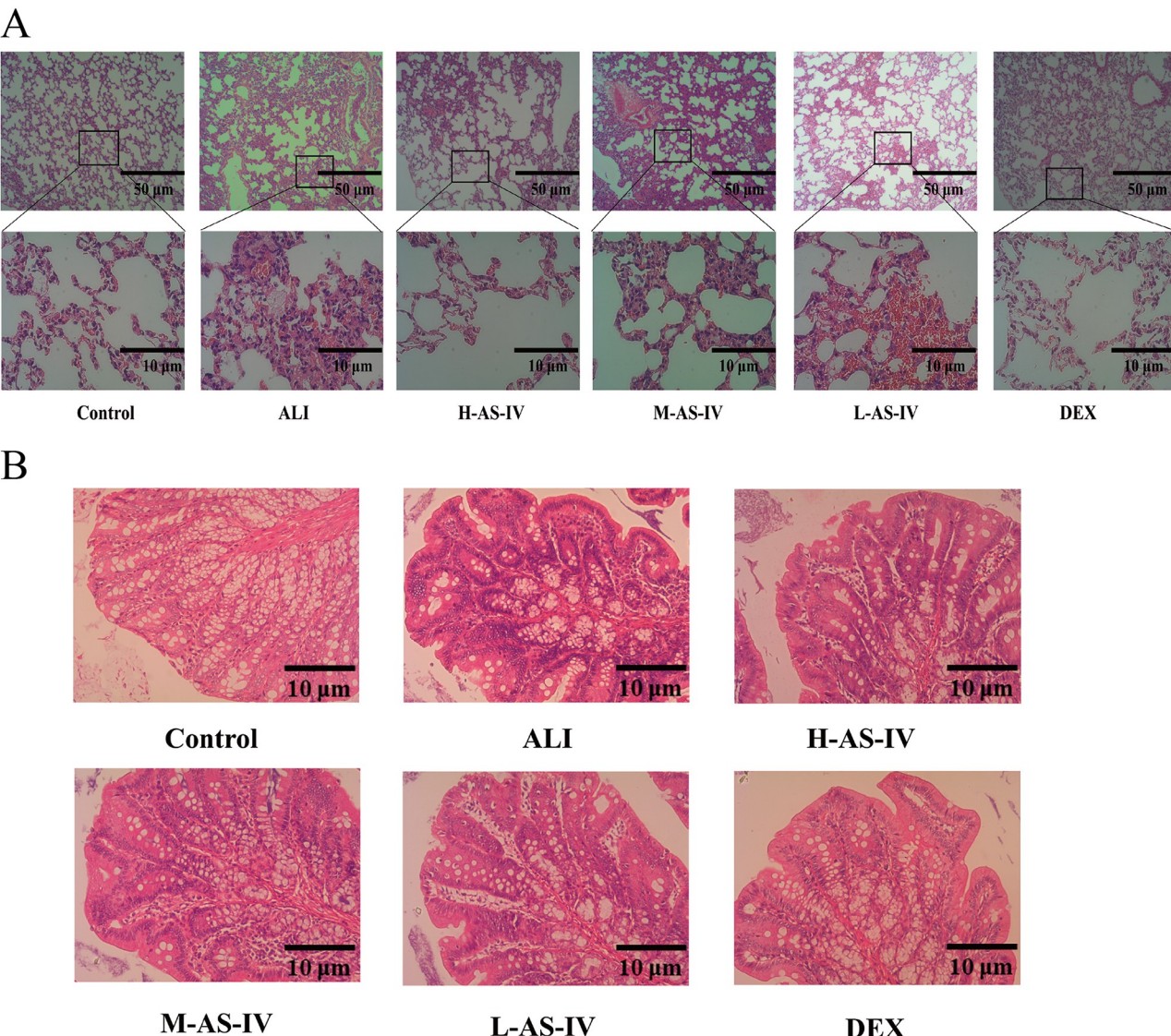

**Fig 2. Demonstrates that AS-IV mitigates the injury to the lungs and colon caused by LPS.** (A) The therapeutic impact of AS-IV in decreasing the severity of lung pathology (Bar = 10μm and 50 μm). (B) The therapeutic impact of AS-IV in mitigating the severity of colon pathology (Bar = 10μm).

M-AS-IV groups were significantly decreased ($P < 0.01$). Serum levels of IL-6 and TNF-α in L-AS-IV and DEX groups were significantly decreased ($P < 0.05$), IL-1β also was decreased, but the difference was not statistically significant ($P > 0.05$). The levels of IL-6, IL-1β and TNF-α in BALF of rats in L-AS-IV and DEX groups were decreased ($P < 0.01$, $P < 0.05$). This experiment explains the role of AS-IV in preventing inflammatory responses in the respiratory system, suggesting that AS-IV can reverse the expression of inflammatory factors.

## AS-IV regulated the mRNA expression levels of PI3K, AKT and mTOR in LPS-induced ALI rats

To further explore the mechanism of AS-IV on LPS-induced ALI rats, we detected the mRNA expression levels of PI3K, AKT and mTOR in lung tissue, as shown in Fig 4A–4C. The study

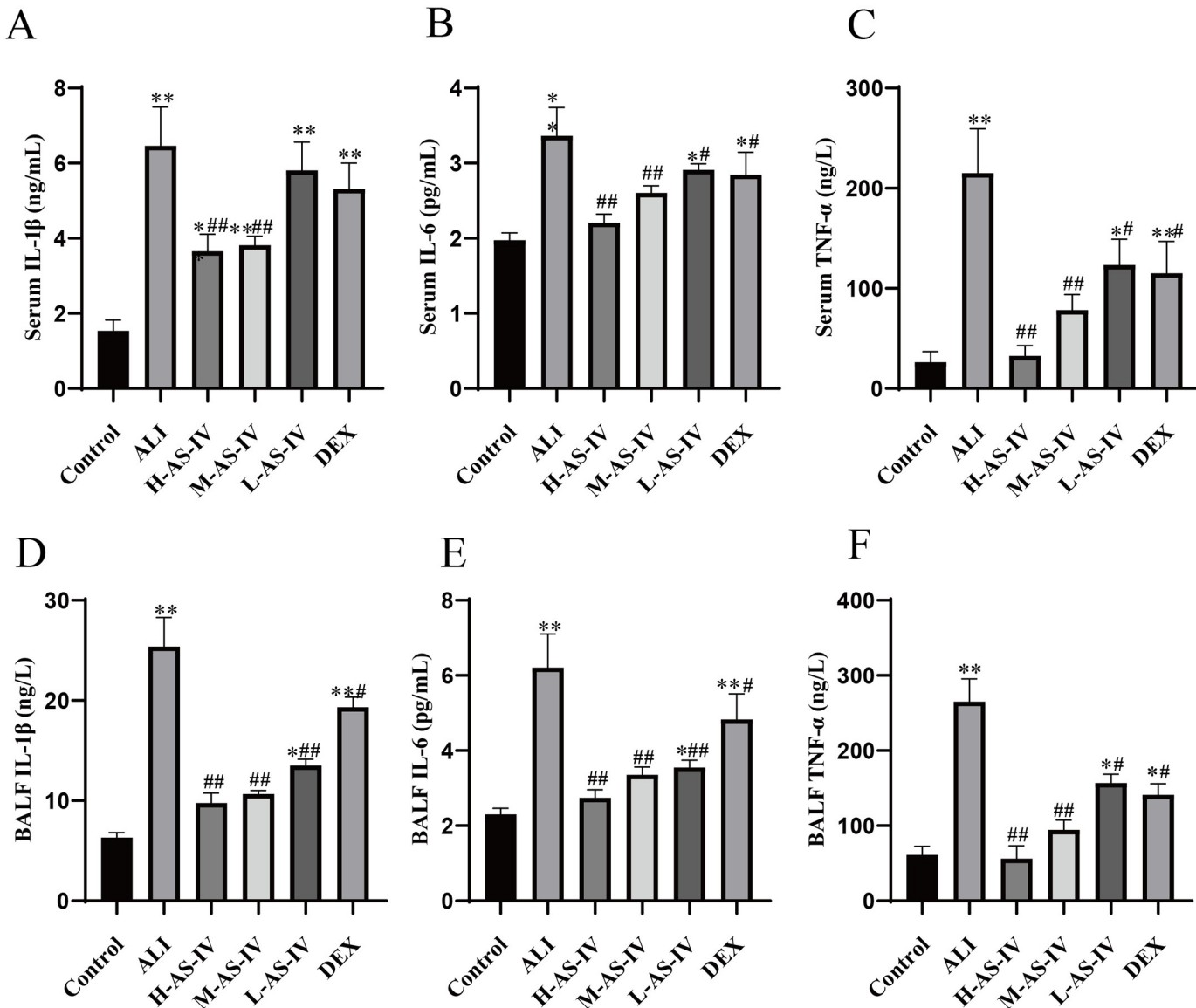

**Fig 3. AS-IV inhibits the expression of inflammatory factors in rats with LPS-induced ALI.** The activity of (A)serum tumor necrosis factor-α (TNF-α), (B) serum Interleukin-1β (IL-1β), (C) serum interleukin-6 (IL-6), (D) BALF tumor necrosis factor-α (TNF-α), (E) BALF Interleukin-1β (IL-1β), and (F) BALF interleukin-6 (IL-6) was evaluated after the administration of AS-IV or DEX to rats. The assessment was conducted using mean ± SEM values from 6 animals in each group. The symbols represent significance levels of $\#P < 0.05$, $\#\#P < 0.01$, $\#\#\#P < 0.001$, respectively.

proved that the mRNA expression levels of PI3K, AKT and mTOR in ALI group were significantly increased ($P < 0.01$). Compared with the ALI group, the expression levels of PI3K and mTOR were significantly downregulated in the AS-IV dose groups and DEX groups ($P < 0.01$), while the expression levels of AKT were significantly downregulated in the H-AS-IV and DEX groups ($P < 0.05$). However, the mRNA expression level of AKT decreased after M-AS-IV and L-AS-IV intervention, but the difference was not statistically significant ($P > 0.05$). After AS-IV treatment, their expression levels were inhibited to varying degrees, and the degree of inhibition was correlated with the dose, that is, the greater the dose of AS-IV, the more obvious the inhibition was. Among them, the reversal of PI3K and mTOR mRNA

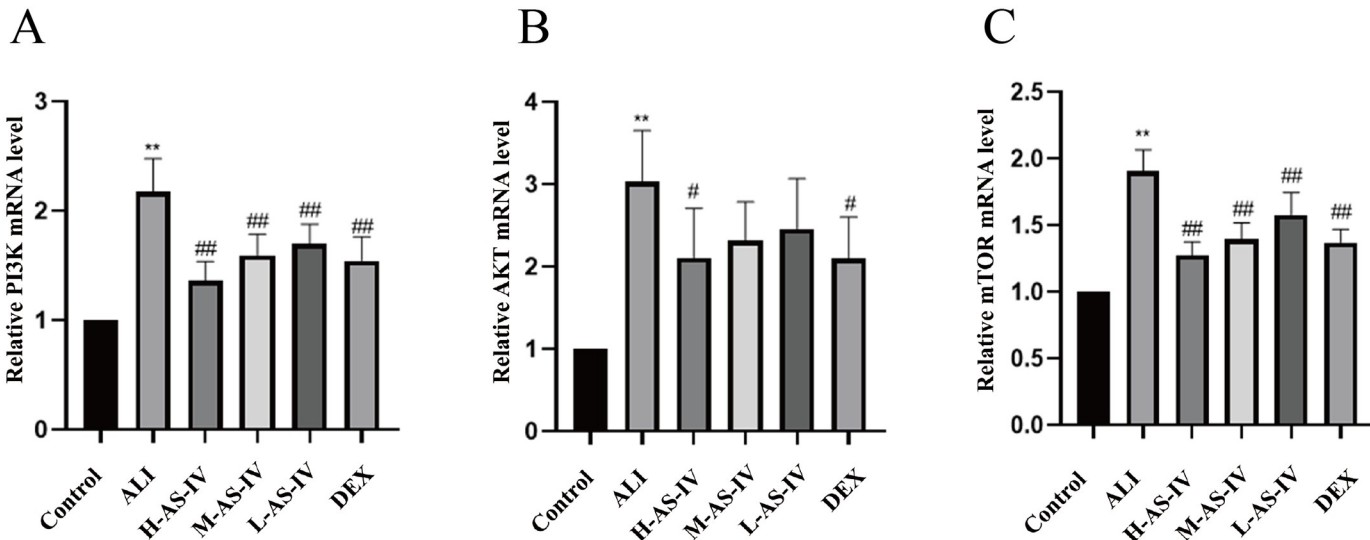

**Fig 4. The impact of AS-IV on the mRNA expression levels of PI3K, AKT, and mTOR in lung tissue in rats with LPS-induced ALI is illustrated.** (A) Effect of AS-IV on PI3K mRNA expression in lung tissue, (B) Effect of AS-IV on AKT mRNA expression in lung tissue, (C) Effect of AS-IV on mTOR mRNA expression in lung tissue. For every group, the mean ± SEM is calculated using data from 6 animals. The symbols have significance levels of #$P < 0.05$, ##$P < 0.01$, ###$P < 0.001$, respectively.

expression levels was most significant in H-AS-IV group ($P < 0.01$). Accordingly, we hypothesized that AS-IV has a regulatory effect on the PI3K/AKT/mTOR pathway in ALI rats and verified it in further experiments.

## AS-IV inhibits the expression of PI3K/AKT/mTOR signaling pathway in LPS-induced ALI rats

To further explore the impact of AS-IV on the PI3K/AKT/mTOR signaling pathway, we detected the protein expression levels associated with the PI3K/AKT/mTOR signaling pathway in lung tissue of rats with LPS-induced ALI (Fig 5A–5F). The results of the experiment indicated that compared with the Control group, the protein expression levels of PI3K, AKT, mTOR, P-PI3K /PI3K, p-Akt /AKT and P-mtor /mTOR in ALI group were significantly up-regulated ($P < 0.01$). Compared with ALI group, the protein expression levels of P-PI3K /PI3K, P-Akt /AKT and P-mTOR /mTOR in AS-IV and DEX groups were significantly down-regulated ($P < 0.01$, $P < 0.05$). The protein expression levels of PI3K, AKT and mTOR were significantly down-regulated in AS-IV groups ($P < 0.01$), the protein expression levels of PI3K and mTOR were significantly down-regulated in DEX group ($P < 0.01$, $P < 0.05$), and the protein expression of AKT was down-regulated, but the differences were not statistically significant ($P > 0.05$). These results showed that we successfully reversed the expression levels of these proteins after treatment with AS-IV, indicating that AS-IV can inhibit the expression of PI3K/AKTmTOR pathway in LPS-induced ALI rat models.

## AS-IV altered gut microbial structural diversity

Studies have shown that ALI is often accompanied by dysregulation of the lung-intestinal axis. Therefore, with the objective learn more about the connection between ALI and gut microbiot. We collected stool samples from each group of rats and performed the 16S rRNA gene sequencing conducted, and we examined how different AS-IV dose groups affected the gut

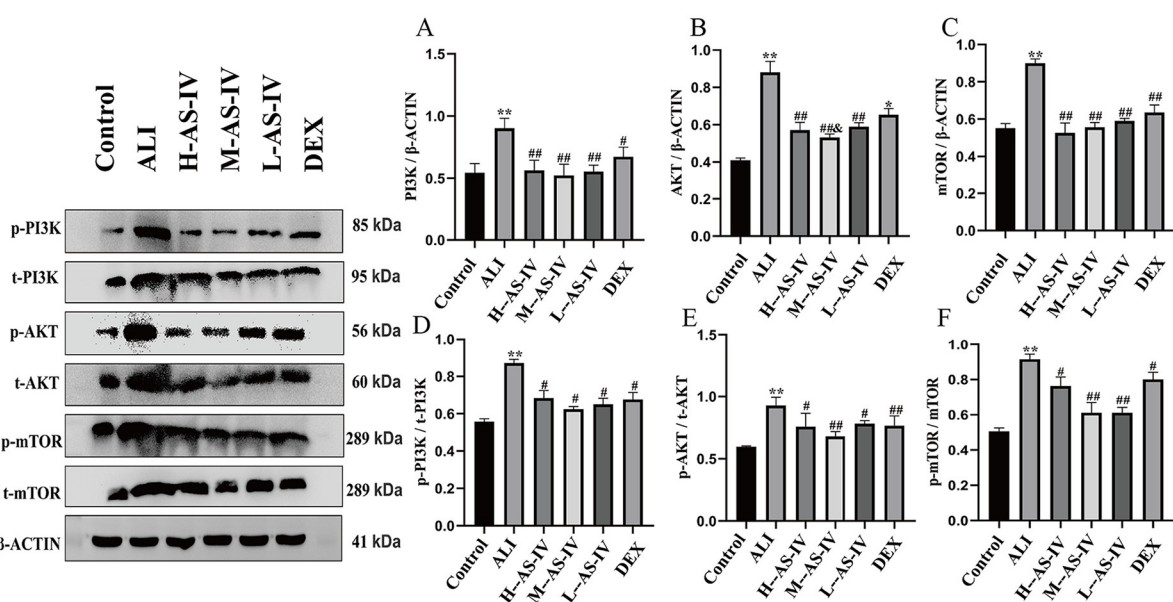

**Fig 5. The influence of AS-IV on PI3K/AKT/mTOR signaling pathway in LPS-induced ALI rats.** After administering AS-IV or DEX, Western blots were performed to analyze the levels of PI3K, AKT, mTOR, p-PI3K/t-PI3K (D), p-AKT/t-AKT (E), and p-mTOR/t-mTOR (F) in the lung tissues of each group rats. β-ACTIN was used to probe the internal reference protein (A-C). For every group, the mean ± SEM is calculated using data from 6 animals. The symbols represent significance levels of #*P< 0.05*, ##*P< 0.01*, ###*P< 0.001*, respectively.

microbiota in ALI rats. The results showed that the dilution curve steadily flattened with increased sequencing times, demonstrating that the sample species diversity was substantial and the data were trustworthy (Fig 6A). α diversity is an index used to evaluate microbial species diversity in a sample, and α diversity index includes Chao1 index, Pielou's evenness index, Observed species index, Shannon index, Good's coverage index and Simpson index, etc. (Fig 6B). Chao1 index and Observed species index are flora abundance index, and the larger the index, the higher the richness of the community. In comparison to the Control group, the ALI group's Chao1 index and Observed species index declined. Compared with ALI group, the Chao1 index and Observed species index in DEX group and AS-IV different dose groups were significantly increased. The Shannon index and Simpson index were microbial diversity indexes, which were positively correlated with the richness and evenness of the community. The higher their values, the more diverse the community. Compared with the Control group, Shannon index in ALI group increased, but Simpson index had no significant change. Compared with ALI group, Shannon index and Simpson index were significantly reduced index group, and increased in AS-IV groups with different doses. Pielou's evenness index is the evenness of the flora. The higher the index is, the more uniform the flora is. In comparison to the Control group, Fig 6B shows: Pielou's evenness index increased in ALI group. Compared with ALI group, Pielou's evenness index was significantly decreased in DEX group and increased in M-AS-IV group, while there was no significant change in L-AS-IV and H-AS-IV groups. The Good's coverage index is to evaluate the coverage of species in the flora. The higher the index is, the smaller the proportion of undetected species in the sample. In all groups, the Good's coverage index is greater than 0.99 and less than 1, indicating that all species in the sample are basically covered.

The gut microbiota of ALI rats was examined using principal coordinate analysis (PCoA) and nonmetric multidimensional scaling analysis (NMDS). The results showed that the PCoA

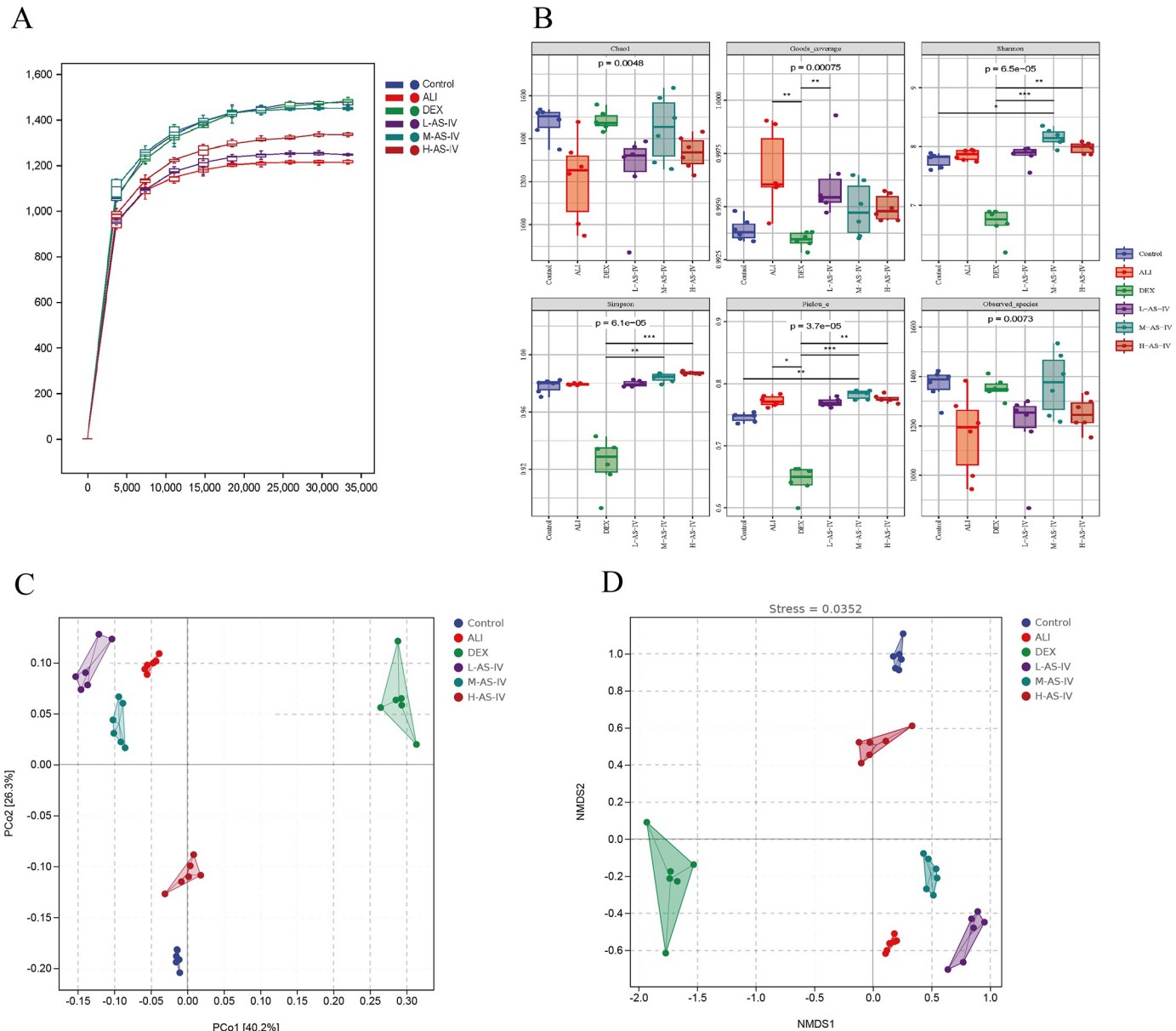

**Fig 6. In LPS-induced ALI rats, the diversity of fecal microbial was changed by AS-IV.** We detected the 16S rRNA sequencing of the fecal samples from the Control group, ALI group, DEX group, and AS-IV low, medium, and high dose group, each consisting of 6 animals. To assess the variety of microorganisms in each sample, a dilution curve (A) is employed. The assessment of α-Diversity involved the utilization of various indices including Chao1, Pielou's evenness, Observed species, Shannon, Good's coverage, and Simpson indices (B) at the operational taxonomic unit (OTU) levels. β-diversity was assessed by performing PCoA analysis (C) and NMDS analysis (D) on the OTU levels.

results varied among the Control group, ALI group, DEX group, and AS-IV groups with different doses (ANOSIM P < 0.01). The findings suggest that treatment with AS-IV has the potential to alter the diversity of intestinal flora in LPS-induced ALI (Fig 6C). The NMDS analysis demonstrates the variation in distance among and within groups, where a shorter (longer) distance between two points signifies a lesser (larger) disparity in microscopic communities between them. Fig 6D illustrates that the Control group and H-AS-IV group are positioned above NMDS2 0.0, while the ALI group, DEX group, L-AS-IV group, and M-AS-IV group are

positioned below NMDS2 0.0, indicating that the gut microbiota was altered by LPS-induced ALI. Nevertheless, the gut microbiome following H-AS-IV therapy exhibited resemblance to the Control group's microbiota. The NMDA stress value of 0.035 (below 0.2) suggests a dependable model. To summarize, AS-IV has the ability to modify the variety of gut microbiota in rats with LPS-induced ALI.

### AS-IV altered fecal microflora composition in LPS-induced ALI rats

To identify variations in particular strains and search for diverse bacteria that could contribute to the emergence of ALI, we explored alterations in the gut microbiota at the phylum and genus tiers. The findings, depicted in Fig 7A, revealed that Firmicutes, Bacteroidetes, Actinobacteria, and Proteobacteria were the prevailing phyla in all six groups. In the ALI and DEX groups, compared to the Control group, the relative abundance of Firmicutes fell while the relative abundance of Bacteroidetes rose; in the L-AS-IV group, compared to the ALI group, Firmicutes had a lower relative abundance while Bacteroidetes had a higher relative abundance; nevertheless, in the M-AS-IV group and H-AS-IV group, Firmicutes had a higher relative abundance while Bacteroidetes had a lower relative abundance. The relative abundance of major flora in the M-AS-IV group was almost consistent with that in the Control group. Fig 7B illustrates that Lactobacillus, Prevotella, Roseburia, and Oscillospira were the dominant genus microbiota in all six groups. The ALI and DEX groups exhibited a rise in the relative abundance of Oscillospira, whereas the relative abundance of Lactobacillus, Roseburia, and Prevotella declined. The M-AS-IV and H-AS-IV groups exhibited a rise in the relative abundance of Oscillospira, Prevotella, and Roseburia, while the L-AS-IV group experienced a decline in the relative abundance of Lactobacillus compared to the ALI group. The relative abundance of major flora in the M-AS-IV group was most similar to that in the Control group. Consequently, AS-IV has the potential to increase the number of beneficial bacteria and reduce the number of harmful bacteria, thereby improving the intestinal flora disturbance.

Furthermore, to distinguish the specific bacteria in each category, a cladogram was generated through the utilization of Linear discriminant analysis Effect Size (LEfSe). By applying a logarithmic LDA score cutoff of 3.5, we were able to pinpoint 40 microorganisms that serve as significant differentiators. Fig 7C and 7D illustrates that f_Lactobacillaceae, g_Lactobacillus, and p_Firmicutes exhibited notable variances among the groups. These taxa were notably more abundant in the Control group, suggesting their substantial impact on the group differences. f_Lachnospiraceae was significantly enriched in ALI group and had the highest abundance, which had a greater effect on the difference between groups. In DEX group, f_Enterococcaceae,f_Enterococcaceae,g_Enerococcus,o_Lactobacillales and c_Bacilli had greater influence on the difference between groups. In AS-IV different dose groups, o_Clostridiaceae and c_Clostridia were significantly enriched and had the highest abundance, which had bigger effects on the difference in groups.

## Discussion

ALI is a prevalent and significant condition in medical practice, often presenting with infiltration of neutrophils in the lungs, damage to endothelial and epithelial cells, disruption of the integrity of the alveolar-capillary membrane, and the release of inflammatory substances [26]. Although there are many treatments available, neither the quality of life nor mortality among those with acute lung injury are improved by them. Increasing amounts of evidence indicate that Chinese medicinal plants and their active components have proven efficacy in the treatment of acute lung injury. This is achieved by diminishing the release of inflammatory mediators, mitigating oxidative stress, and other related mechanisms [27–30]. Recent studies have

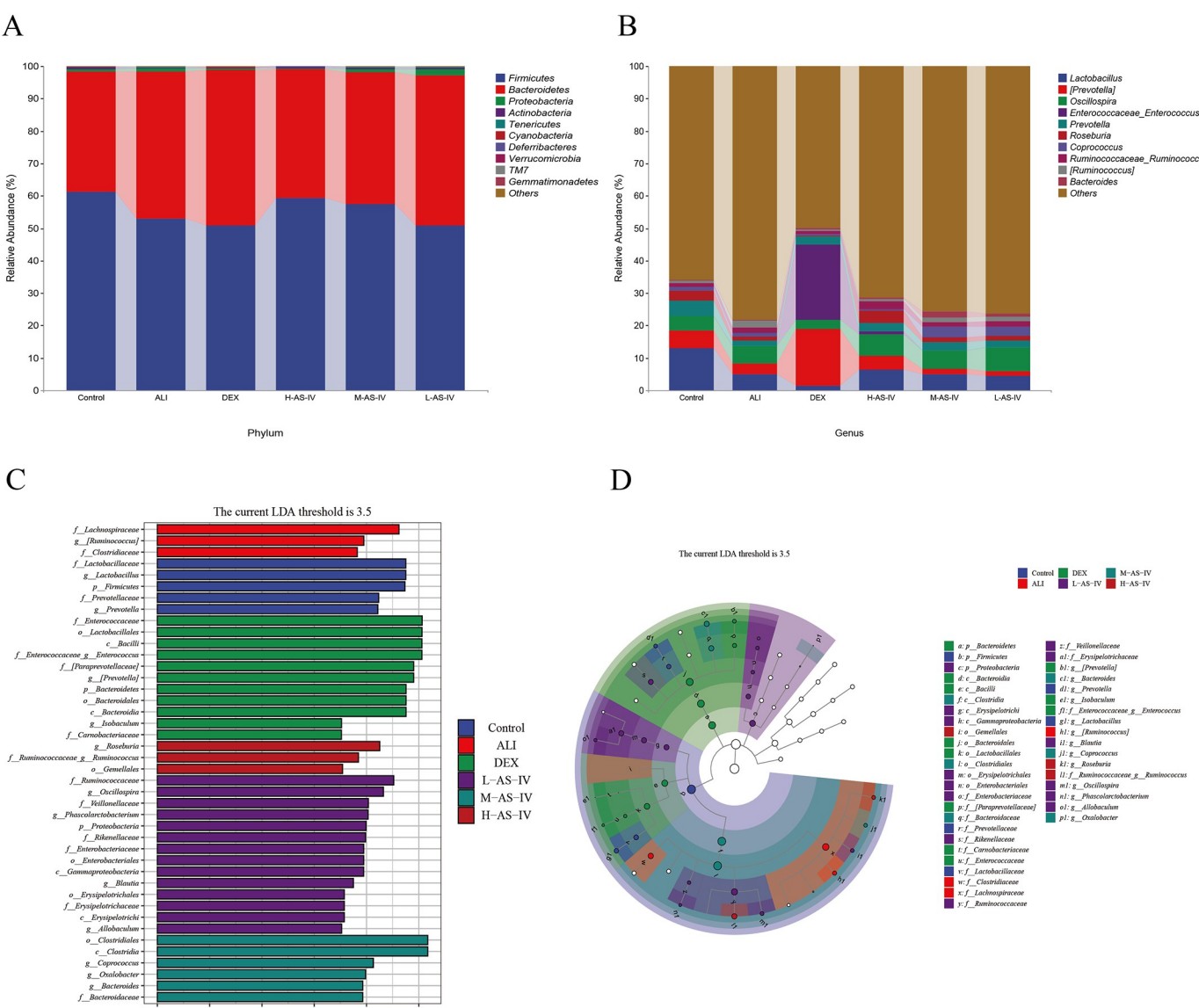

**Fig 7. AS-IV modified the composition of fecal microflora in rats with LPS-induced ALI.** The distribution of community abundance in the Control, ALI, DEX, and AS-IV groups with different doses was analyzed at the phylum (A) and genus (B) levels. The relative abundance (%) of the top 10 phyla and top 10 genera in each group were used to depict the data. Linear discriminant analysis (LDA) was employed to assess the effect size (LEfSe). (C) The variations in abundance among the Control, ALI, DEX, and AS-IV different dose groups.(D) Cladogram showing the microbiota's phylogenetic distribution and how it relates to each group.

found that AS-IV has a protective effect on lung injury caused by pulmonary ischemia-reperfusion [31].

Therefore, the rat model of ALI induced by LPS was chosen to proved the possible mechanism of AS-IV therapy for ALI. ALI typically triggers inflammation, which is marked by elevated levels of inflammatory markers in BALF and serum [32, 33]. Prior research has shown increased levels of IL-6 and IL-1β in both BALF and serum in models of ALI [30]. The research indicated that rats experiencing ALI induced by LPS exhibited notably elevated concentrations of IL-6, TNF-α, and IL-1β in both their BALF and serum. However, these levels decreased significantly after treatment with dexamethasone and AS-IV. In the rat model of LPS-induced

ALI, not only was the alveolar structure disorganized in the lung tissue, but the alveolar space was also widened. Additionally, there was alveolar and interstitial lung edema, bleeding, and a massive infiltration of inflammatory cells. Furthermore, the colon tissue exhibited mucosal hyperemia, structural destruction, and varying degrees of inflammation. The mucosal and sub-mucosal edema and cell infiltration caused a significant amount of inflammation and glandular necrosis. This demonstrates that there is a specific correlation between the pulmonary and intestinal systems, in line with the lung-gut axis suggested by contemporary medical research [34]. The restoration of lung tissue and colon tissue was observed in both the DEX group and AS-IV groups, with H-AS-IV demonstrating the most pronounced impact. Accordingly, we believe that AS-IV possesses the ability to mitigate inflammation in the lungs and colon induced by LPS, while also reducing the release of inflammatory mediators. This suggests that there is a connection between the lungs and intestines.

One of the major intracellular signal transduction pathways, the PI3K/AKT/mTOR signaling system is crucial for cell growth, proliferation, migration, invasion, apoptosis, autophagy, etc. [35–38]. Previous research has demonstrated a correlation between PI3K/AKT/mTOR and cerebral ischemia-reperfusion injury (IRI), indicating that the regulation of the PI3K/AKT/mTOR signaling pathway can potentially prevent IRI by inhibiting apoptosis, autophagy, inflammatory response, and oxidative stress [39–42]. Recent studies have also suggested a potential association between the PI3K/AKT/mTOR signal transduction pathways and acute lung injury, such as controlling the PI3K/AKT/mTOR pathways can diminish inflammation, modulate immune response, inhibit oxidative stress, and alleviate ALI induced by LPS [43–45]. Controlling the PI3K/AKT/mTOR signaling pathways has been demonstrated to decrease inflammation, modulate immune response, hinder oxidative stress, and relieve LPS-induced ALI. The PI3K/AKT/mTOR signaling pathway may be an endogenous negative feedback or compensation mechanism closely related to LPS induced ALI and inflammatory response. A study has found that TLRs initiate a pro-inflammatory signaling cascade, inhibit the activation of PI3K/AKT, reduce cellular inflammatory storms, and alleviate lung injury. In addition, there is also some controversy about PI3K, and some studies have shown that PI3K can promote NF-κB activation induces inflammatory response, while other studies have shown that PI3K can inhibit inflammatory response [46, 47]. To further investigate the potential therapeutic effect of AS-IV in LPS induced ALI rats, we investigated the PI3K/AKT/mTOR signaling pathway. The experiment results indicate that, when compared to the Control group, the ALI group exhibited a significant increase in the mRNA expression levels of PI3K, AKT, and mTOR, as well as the protein expression levels of p-PI3K/t-PI3K, p-AKT/t-AKT, and p-mTOR/t-mTOR. However, the expression of these proteins was reversed in different dosage groups. AS-IV administration markedly decreased the levels of PI3K, AKT, and mTOR proteins. Therefore, targeting the PI3K/AKT/mTOR signaling pathway could be a viable approach for treating LPS-induced ALI in rats, as it helps regulate the inflammatory response caused by LPS. However, there are certain limitations in this experiment. Further verification is needed to determine whether there is a certain relationship between PI3K/AKT/mTOR and the microbiota and its metabolites, and whether the microbiota and its produced metabolites can alleviate lung and colon injury by regulating the PI3K/AKT/mTOR signaling pathway; In addition, the PI3K/AKT/mTOR signaling pathway also has regulatory effects on cell autophagy and apoptosis. This study only investigated its regulation of inflammatory response, which is too limited. Further experiments are needed to explore whether AS-IV can regulate cell autophagy and apoptosis through the PI3K/AKT/mTOR signaling pathway.

The human gastrointestinal tract is inhabited by microbiota, which is referred to as gut microbiota [7]. According to recent research, alterations in the microbiota of the gastrointestinal tract can impact the respiratory system through co-mucosal immunity. Similarly, the

disruption of the respiratory tract microbiota can also influence the gastrointestinal tract through immune regulation. The lung-gut axis, also referred to as the respiratory-digestive tract bidirectional regulation, is facilitated by this microbiome [8, 48, 49]. The intestinal microbiota and its metabolites have been found to be associated with multiple respiratory disorders, as indicated by recent research [50–53]. Acute lung tissue injury is often manifested by a reduction in gut microbial diversity and structural composition [54]. This study utilized the high-throughput sequencing approach of 16S rRNA to investigate the regulatory effect of AS-IV on the intestinal microbiota in ALI rats. The study revealed that the intestinal composition of ALI rats induced by LPS was disrupted, leading to alterations in α diversity. However, treatment with AS-IV was able to reverse the damage to the intestinal structure and restore α diversity. The above findings indicated that AS-IV had the potential to reverse the damage caused by LPS to the intestinal structure and disturbance of intestinal microflora in rats with ALI.

This study found that at the phylum level, Bacteroides and Firmicutes were the dominant bacteria with the highest relative abundance in the intestinal flora. Compared with the Control group, the relative abundance of Firmicutes in the ALI and DEX groups decreased, while the relative abundance of Bacteroides increased. The relative abundance of Firmicutes in M-AS-IV and H-AS-IV groups increased, while the relative abundance of Bacteroides decreased. Some studies have found that some Bacteroides can act on the inflammatory response pathway and aggravate the inflammation of the body [55]. The experiments have proved that Bacteroides has the potential pathogenic effect and can induce the occurrence of colitis [56]. In addition, the lower abundance of Bacteroides is beneficial for the treatment of type 2 diabetes [57]. Most strains of Firmicutes belong to probiotics, which can maintain the homeostasis of intestinal flora. Studies have confirmed that the destruction of Firmicutes will increase the incidence of bacterial wilt disease [58]. Therefore, AS-IV can regulate the levels of Bacteroides and Firmicutes in the intestinal tract of LPS-induced ALI rats. Lactobacillus, Oscillospira, Roseburia, and Prevotella are the dominant bacteria at the genus level. Comparing the ALI and DEX groups to the Control group, Oscillospira's relative abundance rose while Lactobacillus, Roseburia, and Prevotella's relative abundance decreased. When compared to the ALI group, the relative abundances of Oscillospira, Prevotella, and Roseburia increased in the L-AS-IV group, while Lactobacillus declined. In comparison to the ALI group, the L-AS-IV group exhibited higher levels of Oscillospira, Prevotella, and Roseburia, whereas Lactobacillus experienced a decrease. In the M-AS-IV and H-AS-IV groups, there was an increase in the proportions of Lactobacillus, Oscillospira, Roseburia, and Prevotella. It should be mentioned that Prevotella is recognized as a probiotic, and scientific studies have demonstrated that decreasing Prevotella levels can lead to a reduction in intestinal harm and inflammation [59]. Lactobacillus is a potential probiotic that activates the biosynthesis of primary bile acids in the liver and promotes the biotransformation of secondary bile acids in the gut [60]. When there is an infection or inflammation in the body, the abundance of gut Lactobacillus decreases [61]. Studies have found that Oscillospira is related to inflammation to some extent, and the intestinal Oscillospira of patients with inflammation is less than that of normal people [59, 62]. Roseburia is considered to be a probiotic that has a significant effect on improving intestinal diseases [63]. Studies have shown that increases in intestinal Eubacterium and Roseburia inhibit the growth of colon cancer [64]. In short, these results provide strong evidence for a significant link between changes in gut microbiota and body diseases, especially inflammatory diseases. In our study, DEX, although effective in reversing the expression of inflammatory factors, was not effective in optimizing the structure of intestinal microorganisms. AS-IV was effective in reducing the expression of inflammatory factors and optimizing the structure of

the intestinal flora in rats with ALI. However, the mechanism by which AS-IV regulates the intestinal flora and its metabolites to produce effects on the lungs needs to be further explored.

Through the above discussion, we found that AS-IV not only improves lung injury, but also has a certain impact on gut microbiota. In pathological sections, it was observed that AS-IV treatment can effectively improve the damaged structure of lung and colon tissues. Its mechanism of action is to inhibit the PI3K/AKT/mTOR signaling pathway, reduce inflammatory cell infiltration and cytokine release, and regulate the diversity of gut microbiota, increase the abundance of beneficial bacteria in the gut microbiota, and reduce the abundance of harmful bacteria. This confirms the correlation between lungs and intestines, which may be achieved through the "gut-lung" axis. This experiment suggests that AS-IV may regulate the lung and gut microbiota of ALI rats through the "gut-lung" axis, reduce inflammatory factor levels, and improve lung and colon injury, which is related to the inhibition of PI3K/AKT/mTOR signaling pathway expression. And we speculate that this effect may be related to the transfer of microbiota between the lungs and intestines and the interaction of metabolites, which requires further experimental verification.

## Conclusion

Our study confirmed the specific therapeutic effect of AS-IV on LPS-induced ALI rats. AS-IV attenuated lung and colon tissue damage and reversed the expression of inflammatory factors in LPS-induced ALI rats, which may be related to the PI3K/AKT/mTOR pathway. In addition, he could rebuild the intestinal microbiota and fulfill its potential role in the treatment of ALI. In conclusion, AS-IV may be a potentially effective and relatively safe drug for the treatment of ALI, which provides us with a choice for clinical treatment of ALI and is worthy of further study. And in the future we hope that further research can be conducted on the transfer of bacterial communities between the lungs and intestines, as well as the interaction of their related metabolites.

## Supporting information

**S1 Graphical abstract.**
(DOCX)

**S2 Graphical abstract.**
(TIFF)

**S1 Raw images.**
(DOCX)

**S1 Data.**
(ZIP)

## Acknowledgments

We are grateful for Yuanhang Ye's proofreading of the article. We thank every member of the team for their contribution. Thank Figdraw for providing the drawing software.

## Author Contributions

**Conceptualization:** Cheng Luo, Yuanhang Ye.

**Data curation:** Cheng Luo, Yuanhang Ye.

**Formal analysis:** Anqi Lv, Wanzhao Zuo.

**Funding acquisition:** Yi Yang, Jia Ke.

**Methodology:** Cheng Luo.

**Project administration:** Jia Ke.

**Software:** Cheng Luo, Cheng Jiang.

**Validation:** Cheng Luo, Cheng Jiang.

**Visualization:** Cheng Luo, Cheng Jiang.

**Writing – original draft:** Cheng Luo.

**Writing – review & editing:** Cheng Luo, Yuanhang Ye.

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
