## [Decision Letter · Decision Letter 0]

5 Mar 2024

PONE-D-23-42843The impact of Astragaloside IV on the inflammatory response and Gut microbiota in cases of acute lung injury is examined through the utilization of the PI3K/AKT/mTOR pathway.PLOS ONE

Dear Dr. Ke,

Thank you for submitting your manuscript to PLOS ONE. After careful consideration, we feel that it has merit but does not fully meet PLOS ONE’s publication criteria as it currently stands. Therefore, we invite you to submit a revised version of the manuscript that addresses the points raised during the review process.

We look forward to receiving your revised manuscript.

Kind regards,

Palash Mandal

Academic Editor

PLOS ONE

Journal Requirements:

"This work was supported by the construction project of Inheritance Studio of the State Administration of Traditional Chinese Medicine [grant number Teaching Letter 2022 No. 245]; Hubei Province "Public Health Youth Top Talent Training Program" project [grant number E Weitong 2021 74]"

**Additional Editor Comments:**

Dear Authors,

Thank you for submitting your manuscript to PLOS ONE. After careful consideration, we feel that it has merit but does not fully meet PLOS ONE publication criteria as it currently stands. The shortcomings of this paper needs to be worked out before it can be considered for publication. Therefore, we invite you to resubmit a revised version of the manuscript that addresses the points raised during the review process.

For your guidance, the reviewers' comments are included below.

Thank you for giving us the opportunity to consider your work.

Specific concerns expressed during peer review were:

Editor’s comments:

This manuscript addresses a timely topic and makes a relevant contribution to the field which elaborates how AS-IV reduces the expression of inflammatory factors by inhibiting the PI3K/AKT/mTOR pathway and optimizes the composition of the gut microflora in AIL rats. The author could have been explored the possibility of potential involvement of other potential molecular mechanistic pathways like MAPK, ERK or JAK-STAT pathways.

Comments from Reviewer 1

Dear Authors,

Your manuscript titled "The impact of Astragaloside IV on the inflammatory response and Gut microbiota in cases of acute lung injury examined through the PI3K/AKT/mTOR pathway" presents a comprehensive study exploring the therapeutic potential of Astragaloside IV (AS-IV) in the context of acute lung injury (ALI) and its effect on gut microbiota through modulation of the PI3K/AKT/mTOR pathway. The study is well-structured, and the methodology is robust, covering a wide range of techniques from histological analysis to gene expression and microbiota composition analysis. However, there are several areas where improvements could enhance the clarity and impact of your findings:

1. Literature Review: While your introduction provides a good background, it would be beneficial to further discuss previous studies that have explored similar treatments for ALI, particularly focusing on those involving gut microbiota modulation. This would contextualize your findings within the broader research landscape.

2. Methodological Details: In several sections, the methodology could benefit from more detailed descriptions to ensure reproducibility. For example, specifics regarding the conditions under which the 16S rRNA sequencing was performed (such as sequencing depth and bioinformatics tools used) would be valuable.

3. Statistical Analysis: While you have employed appropriate statistical tests, the manuscript would benefit from a more detailed explanation of the choice of these tests and any assumptions made. Additionally, include information on the corrections applied for multiple comparisons, if any.

4. Results Interpretation: The results are compelling but could be further strengthened by a more detailed discussion of the implications of your findings, particularly the mechanistic insights into how AS-IV modulates the gut-lung axis. Consider discussing potential limitations of the PI3K/AKT/mTOR pathway's role in mediating these effects.

5. Figures and Tables: Ensure that all figures and tables are clearly labeled and referenced in the text. Some figures, such as the graphical abstract and histological images, could benefit from higher resolution or more detailed annotations to aid interpretation.

6. Conclusion and Future Directions: The conclusion succinctly summarizes the findings but could be expanded to more explicitly state the potential clinical implications of your research and suggest specific areas for future studies.

7. References: Check the currentness of your references, especially focusing on recent advances that might complement your discussion.

8. Ethical Considerations: Your ethical approval statement is in order, but please ensure that any additional ethical considerations, particularly concerning animal welfare, are comprehensively addressed.

Comments from Reviewer 2

The present manuscript entitled 'The impact of Astragaloside IV on the inflammatory response and Gut microbiota in

cases of acute lung injury is examined through the utilization of the PI3K/AKT/mTOR

pathway.' describes efficacy of a natural triterpenoid saponin compound, Astragaloside IV(AS-IV), in the treatment of AIL by the suppression of the inflammatory factors through

PI3K/AKT/mTOR pathway and optimizes the composition of the gut microflora in AIL

rats. The study has several shortcomings though is of scientific merit.

Q1. What was the rationale of selecting the dose and duration of AS-IV? What was the route of injection of AS-IV?

Q2. What is the rationale for selecting PI3K/AKT/mTOR pathway? Is it just because as they mentioned 'Additionally, AV-IV has demonstrated its ability to safeguard against lung injury caused by PM2.5 by activating the PI3K/AKT/mTOR signaling pathway'?

They need to prove the involvement of these pathways further by using some inhibitor (of these pathways) study in vitro.

Q3. What about the toxicity if any of usage of AS-IV?

Reviewers' comments:

Reviewer's Responses to Questions

**Comments to the Author**

1. Is the manuscript technically sound, and do the data support the conclusions?

Reviewer #1: Yes

Reviewer #2: Partly

2. Has the statistical analysis been performed appropriately and rigorously? 

Reviewer #1: Yes

Reviewer #2: Yes

3. Have the authors made all data underlying the findings in their manuscript fully available?

Reviewer #1: Yes

Reviewer #2: Yes

4. Is the manuscript presented in an intelligible fashion and written in standard English?

Reviewer #1: Yes

Reviewer #2: Yes

5. Review Comments to the Author

Reviewer #1: Dear Authors,

Your manuscript titled "The impact of Astragaloside IV on the inflammatory response and Gut microbiota in cases of acute lung injury examined through the PI3K/AKT/mTOR pathway" presents a comprehensive study exploring the therapeutic potential of Astragaloside IV (AS-IV) in the context of acute lung injury (ALI) and its effect on gut microbiota through modulation of the PI3K/AKT/mTOR pathway. The study is well-structured, and the methodology is robust, covering a wide range of techniques from histological analysis to gene expression and microbiota composition analysis. However, there are several areas where improvements could enhance the clarity and impact of your findings:

1. Literature Review: While your introduction provides a good background, it would be beneficial to further discuss previous studies that have explored similar treatments for ALI, particularly focusing on those involving gut microbiota modulation. This would contextualize your findings within the broader research landscape.

2. Methodological Details: In several sections, the methodology could benefit from more detailed descriptions to ensure reproducibility. For example, specifics regarding the conditions under which the 16S rRNA sequencing was performed (such as sequencing depth and bioinformatics tools used) would be valuable.

3. Statistical Analysis: While you have employed appropriate statistical tests, the manuscript would benefit from a more detailed explanation of the choice of these tests and any assumptions made. Additionally, include information on the corrections applied for multiple comparisons, if any.

4. Results Interpretation: The results are compelling but could be further strengthened by a more detailed discussion of the implications of your findings, particularly the mechanistic insights into how AS-IV modulates the gut-lung axis. Consider discussing potential limitations of the PI3K/AKT/mTOR pathway's role in mediating these effects.

5. Figures and Tables: Ensure that all figures and tables are clearly labeled and referenced in the text. Some figures, such as the graphical abstract and histological images, could benefit from higher resolution or more detailed annotations to aid interpretation.

6. Conclusion and Future Directions: The conclusion succinctly summarizes the findings but could be expanded to more explicitly state the potential clinical implications of your research and suggest specific areas for future studies.

7. References: Check the currentness of your references, especially focusing on recent advances that might complement your discussion.

8. Ethical Considerations: Your ethical approval statement is in order, but please ensure that any additional ethical considerations, particularly concerning animal welfare, are comprehensively addressed.

Reviewer #2: The present manuscript entitled 'The impact of Astragaloside IV on the inflammatory response and Gut microbiota in

cases of acute lung injury is examined through the utilization of the PI3K/AKT/mTOR

pathway.' describes efficacy of a natural triterpenoid saponin compound, Astragaloside IV(AS-IV), in the treatment of AIL by the supression of the inflammatory factors through

PI3K/AKT/mTOR pathway and optimizes the composition of the gut microflora in AIL

rats. The study has several shortcoming though is of scientific merit.

Q1. What was the rationale of selecting the dose and duration of AS-IV? What was the route of injection of AS-IV?

Q2. What is the rationale for selecting PI3K/AKT/mTOR pathway? Is it just because as they mentioned 'Additionally, AV-IV

has demonstrated its ability to safeguard against lung injury caused by PM2.5 by activating the

PI3K/AKT/mTOR signaling pathway'?

They need to prove the involvement of these pathways further by using some inhibitor (of these pathways) study in vitro.

Q3. What about the toxicity if any of usage of AS-IV?

6. PLOS authors have the option to publish the peer review history of their article (what does this mean?). If published, this will include your full peer review and any attached files.

Reviewer #1: **Yes: **Dr. Prasad Andhare

Reviewer #2: No

---

## [Author Response · Author response to Decision Letter 0]

6 Apr 2024

To Editor：

Thank you for your letter and the constructive comments on this article in your busy schedule. All of us authors have carefully read the comments that you have given us, and have discussed and revised each of these issues. The following is my list of revisions. In addition, we resubmitted two new revised manuscripts, one is an unmarked version of the revised paper, and the other is in revision mode with the revised part marked in red. If there are any incorrect answers or questions in the manuscript, please do not hesitate to let us know.

Respond to the Journal Requirements

We have made some changes to the manuscript as requested by PLOS ONE, but these changes will not affect the content and framework of the paper.

We consider this to be a very valuable suggestion. We have added some content, including: 1. the rats were first anesthetized with isoflurane gas to relieve pain or discomfort. 2. All rats were euthanized and place the rat separately in a small room and gradually add a lethal dose of CO2 concentration (e.g., 33% administered after 1 minute) to make them fall asleep. 3. Let them die by cervical dislocation.

We are sorry for our carelessness. It has been revised in our resubmitted manuscript. Thank you for your correction.

4. Please state what role the funders took in the study. 

We have explained the role of the sponsor in this manuscript and revised it in the cover letter：Jia Ke contributed to conceptualization, study design and decision to publis, but the funders otherwise had no role in data collection and analysis. In addition, research data of this study can be obtained, including blot image data.

5. PLOS requires an ORCID iD for the corresponding author in Editorial Manager on papers submitted after December 6th, 2016.

Thank you very much for your suggestion, the corresponding author in this manuscript has added the ORCID ID.

6. PLOS ONE now requires that authors provide the original uncropped and unadjusted images underlying all blot or gel results reported in a submission’s figures or Supporting Information files

We have made the original uncropped and unadjusted images of all imprinted or gel result reports in PDF format and placed them in Support Information.

7. Please include captions for your Supporting Information files at the end of your manuscript, and update any in-text citations to match accordingly.

We have added captions for your Supporting Information files at the end of your manuscript, and update any in-text citations to match accordingly.

8. PLOS authors have the option to publish the peer review history of their article (what does this mean?). If published, this will include your full peer review and any attached files.

YES

Respond to the Review 1’ comments

1. Literature Review: While your introduction provides a good background, it would be beneficial to further discuss previous studies that have explored similar treatments for ALI, particularly focusing on those involving gut microbiota modulation. This would contextualize your findings within the broader research landscape.

We greatly appreciate your professional comments on our articles. We have carefully read a lot of relevant literature on acute lung injury and intestinal flora, and added more references on acute lung injury and intestinal flora to the introduction of the revised manuscript to enrich the correlation between ALI and intestinal flora and improve the background, which has been marked in red. See the introduction for details, and the specific literature is as follows: (Zhang Z, Chakawa MB, Galeas-Pena M, Frydman JA, Allen MJ, Jones M, et al. IL-22 Binding Protein Controls IL-22-Driven Bleomycin-Induced Lung Injury. Am J Pathol. 2024 Mar; 194(3):338-352.)、(Shi Q, Zeng S, Yu R, Li M, Shen C, Zhang X, et al. The small RNA PrrH aggravates Pseudomonas aeruginosa-induced acute lung injury by regulating the type III secretion system activator ExsA. Microbiol Spectr. 2024 Mar 5; 12(3):e0062623.)、(Dickson RP, Singer BH, Newstead MW, Falkowski NR, Erb-Downward JR, Standiford TJ, et al. Enrichment of the lung microbiome with gut bacteria in sepsis and the acute respiratory distress syndrome. Nat Microbiol. 2016 Jul 18; 1(10):16113.)、(Yoon YM, Hrusch CL, Fei N, Barron GM, Mills KAM, Hollinger MK, et al. Gut microbiota modulates bleomycin-induced acute lung injury response in mice. Respir Res. 2022 Dec 10; 23(1):337.)、(Panzer AR, Lynch SV, Langelier C, Christie JD, McCauley K, Nelson M, et al. Lung Microbiota Is Related to Smoking Status and to Development of Acute Respiratory Distress Syndrome in Critically Ill Trauma Patients. Am J Respir Crit Care Med. 2018 Mar 1; 197(5):621-631.)、(Kyo M, Nishioka K, Nakaya T, Kida Y, Tanabe Y, Ohshimo S, et al. Unique patterns of lower respiratory tract microbiota are associated with inflammation and hospital mortality in acute respiratory distress syndrome. Respir Res. 2019 Nov 6; 20(1):246.).

2. Methodological Details: In several sections, the methodology could benefit from more detailed descriptions to ensure reproducibility. For example, specifics regarding the conditions under which the 16S rRNA sequencing was performed (such as sequencing depth and bioinformatics tools used) would be valuable.

Thank you for your advice. We have perfected animal modeling and treatment, including animal sacrifice, methods of anesthesia and pain relief, as well as the route of AS-IV administration On page 4, lines 19-32. In addition, we also follow your suggestion on the specific conditions for 16S rRNA sequencing as follows: Fecal samples were quickly collected from rat cecum and frozen with liquid nitrogen and stored at -80℃. DNA extraction and 16S rRNA gene sequencing of fecal samples were conducted with the assistance of Wuhan Billion Huineng Biotechnology Co., Ltd. (China). Total genomic DNA samples were extracted using the OMEGA Soil DNA Kit (M5635-02) (Omega Bio-Tek, Norcross, GA, USA), following the manufacturer’s instructions, and stored at -20 °C prior to further analysis. The quantity and quality of extracted DNAs were measured using a NanoDrop NC2000 spectrophotometer (Thermo Fisher Scientific, Waltham, MA, USA) and agarose gel electrophoresis, respectively. Specific amplification of the 16S rRNA gene was performed. PCR amplification of the bacterial 16S rRNA genes V3–V4 region was performed using the forward primer 338F (5'-ACTCCTACGGGAGGCAGCA-3') and the reverse primer 806R (5'-GGACTACHVGGGTWTCTAAT-3'). PCR amplicons were purified with Vazyme VAHTSTM DNA Clean Beads (Vazyme, Nanjing, China) and quantified using the Quant-iT PicoGreen dsDNA Assay Kit (Invitrogen, arlsbad, CA, USA). After the individual quantification step, amplicons were pooled in equal amounts, and pair-end 2 250 bp sequencing was performed using the Illlumina MiSeq platform with MiSeq Reagent Kit v3 at Bioyi Biotechnology Co.,Ltd.(Wuhan, China). It's on page 6, lines 6-21.

3. Statistical Analysis: While you have employed appropriate statistical tests, the manuscript would benefit from a more detailed explanation of the choice of these tests and any assumptions made. Additionally, include information on the corrections applied for multiple comparisons, if any.

We sincerely appreciate your valuable advice. We carefully revised the content of statistical correlation in the paper to provide more detailed explanations, such as the expression levels of inflammatory factors in serum and BALF, PI3K/AKT/mTOR signaling pathway related proteins and m RNA.

On page 7, lines 1-5 from the bottom, and on page 8, lines 1-3 of the text：The results showed that compared with the Control group, the levels of inflammatory factors TNF-α, IL-6 and IL-1β in serum and BALF of rats in ALI group were significantly increased, with statistical significance (P < 0.01). Compared with ALI group, the levels of IL-6, IL-1β and TNF-α in serum and BALF of rats in H-AS-IV and M-AS-IV groups were significantly decreased (P < 0.01). Serum levels of IL-6 and TNF-α in L-AS-IV and DEX groups were significantly decreased (P < 0.05), IL-1β also was decreased, but the difference was not statistically significant (P > 0.05). The levels of IL-6, IL-1β and TNF-α in BALF of rats in L-AS-IV and DEX groups were decreased (P < 0.01, P < 0.05). 

On page 8, lines 1-4 from the bottom, and on page 9, lines 1-7 of the text：The study proved that the mRNA expression levels of PI3K, AKT and mTOR in ALI group were significantly increased (P < 0.01). Compared with the ALI group, the expression levels of PI3K and mTOR were significantly downregulated in the AS-IV dose groups and DEX groups (P < 0.01), while the expression levels of AKT were significantly downregulated in the H-AS-IV and DEX groups (P < 0.05). However, the mRNA expression level of AKT decreased after M-AS-IV and L-AS-IV intervention, but the difference was not statistically significant (P > 0.05). After AS-IV treatment, their expression levels were inhibited to varying degrees, and the degree of inhibition was correlated with the dose, that is, the greater the dose of AS-IV, the more obvious the inhibition was. Among them, the reversal of PI3K and mTOR mRNA expression levels was most significant in H-AS-IV group (P < 0.01). 

On page 9, lines 4-12 from the bottom of the text：The results of the experiment indicated that compared with the Control group, the protein expression levels of PI3K, AKT, mTOR, P-PI3K /PI3K, p-Akt /AKT and P-mtor /mTOR in ALI group were significantly up-regulated (P < 0.01). Compared with ALI group, the protein expression levels of P-PI3K /PI3K, P-Akt /AKT and P-mTOR /mTOR in AS-IV and DEX groups were significantly down-regulated (P < 0.01, P < 0.05). The protein expression levels of PI3K, AKT and mTOR were significantly down-regulated in AS-IV groups (P < 0.01), the protein expression levels of PI3K and mTOR were significantly down-regulated in DEX group (P < 0.01, P < 0.05), and the protein expression of AKT was down-regulated, but the differences were not statistically significant (P > 0.05).

4. Results Interpretation: The results are compelling but could be further strengthened by a more detailed discussion of the implications of your findings, particularly the mechanistic insights into how AS-IV modulates the gut-lung axis. Consider discussing potential limitations of the PI3K/AKT/mTOR pathway's role in mediating these effects.

We think this is a good suggestion. We further explained in our discussion the mechanistic insights on how AS-IV regulates the gut-lung axis on page 16, lines 16-27 of the text, as follows: we found that AS-IV not only improves lung injury, but also has a certain impact on gut microbiota. In pathological sections, it was observed that AS-IV treatment can effectively improve the damaged structure of lung and colon tissues. Its mechanism of action is to inhibit the PI3K/AKT/mTOR signaling pathway, reduce inflammatory cell infiltration and cytokine release, and regulate the diversity of gut microbiota, increase the abundance of beneficial bacteria in the gut microbiota, and reduce the abundance of harmful bacteria. This confirms the correlation between lungs and intestines, which may be achieved through the "gut-lung" axis. This experiment suggests that AS-IV may regulate the lung and gut microbiota of ALI rats through the "gut-lung" axis, reduce inflammatory factor levels, and improve lung and colon injury, which is related to the inhibition of PI3K/AKT/mTOR signaling pathway expression. And we speculate that this effect may be related to the transfer of microbiota between the lungs and intestines and the interaction of metabolites, which requires further experimental verification. However, there are certain limitations in this experiment on page 15, lines 1-8 of the text, highlighted in red. Further verification is needed to determine whether there is a certain relationship between PI3K/AKT/mTOR and the microbiota and its metabolites, and whether the microbiota and its produced metabolites can alleviate lung and colon injury by regulating the PI3K/AKT/mTOR signaling pathway; In addition, the PI3K/AKT/mTOR signaling pathway also has regulatory effects on cell autophagy and apoptosis. This study only investigated its regulation of inflammatory response, which is too limited. Further experiments are needed to explore whether AS-IV can regulate cell autophagy and apoptosis through the PI3K/AKT/mTOR signaling pathway.

5. Figures and Tables: Ensure that all figures and tables are clearly labeled and referenced in the text. Some figures, such as the graphical abstract and histological images, could benefit from higher resolution or more detailed annotations to aid interpretation.

As suggested by the reviewer, all images and tables are clearly marked and referenced in the text, and all high-definition images are uploaded to the attachment.

6. Conclusion and Future Directions: The conclusion succinctly summarizes the findings but could be expanded to more explicitly state the potential clinical implications of your research and suggest specific areas for future studies.

These comments are very valuable and helpful. We have carefully read the relevant opinions and made revisions. In the conclusion section on page 16 of the article, the specific content is as follows: In this experiment, AS AS-IV has certain efficacy in the treatment of ALI, it is effective in regulating intestinal flora and alleviating inflammatory response. Due to the relatively safe and reliable effect of AS-IV, it provides us with a potential choice for future clinical treatment of ALI, which is worthy of further study. And in the future we hope that further research can be conducted on the transfer of bacterial communities between the lungs and intestines, as well as the interaction of their related metabolites.

7. References: Check the currentness of your references, especially focusing on recent advances that might complement your discussion.

Thank you very much for your advice. According to your suggestions, we have added some contents to the discussion, which are on pages 14-16 of the paper.

8. Ethical Considerations: Your ethical approval statement is in order, but please ensure that any additional ethical considerations, particularly concerning animal welfare, are comprehensively addressed.

This study was approved by the Experimental Animal Ethics Association of Hubei University of Traditional Chinese Medicine (HUCMS00302536). And in terms of animal experiments, we took the following measures: during the ALI tracheal instillation modeling process, we first anesthetized rats with isoflurane gas to alleviate pain or discomfort, and then euthanized them. The euthanasia standard was to place the rats separately in a small room and gradually add a lethal dose of carbon dioxide concentration (e.g. 33% after 1 minute) to make them fall asleep. Then, let them die from cervical dislocation.

Respond to the Review 2’ comments

1. What was the rationale of selecting the dose and duration of AS-IV? What was the route of injection of AS-IV?

1.We sincerely accept the professional suggestions of the reviewers on our paper. We choose the dosage and duration of AS-IV mainly for the following reasons: 1. Based on reading relevant literature on the application of AS-IV (Pei, Caixia et al. "Astragaloside IV Protects from PM2.5-Induced Lung Injury by Regulating Autophagy via Inhibition of PI3K/Akt/mTOR Signaling in vivo and in vitro. "Journal of inflammation research vol. 14 4707-4721.16 Sep. 2021), (Li, Min et al. “Astragaloside IV protects against focal cerebral ischemia/reperfusion injury correlating to suppression of neutrophils adhesion-related molecules. "Neurochemistry international vol. 60,5 (2012): 458-65.), (Zhang X, Qu H, Yang T, Liu Q, Zhou H. Astragaloside IV attenu

---

## [Decision Letter · Decision Letter 1]

23 May 2024

The impact of Astragaloside IV on the inflammatory response and Gut microbiota in cases of acute lung injury is examined through the utilization of the PI3K/AKT/mTOR pathway.

PONE-D-23-42843R1

Dear Dr. Jia Ke,

We’re pleased to inform you that your manuscript has been judged scientifically suitable for publication and will be formally accepted for publication once it meets all outstanding technical requirements.

Kind regards,

Palash Mandal

Academic Editor

PLOS ONE